# Factors Associated with Refractive Prediction Error after Phacotrabeculectomy

**DOI:** 10.3390/jcm12175706

**Published:** 2023-09-01

**Authors:** Jung Hye Shin, Seok Hwan Kim, Sohee Oh, Kyoung Min Lee

**Affiliations:** 1Department of Ophthalmology, Seoul National University College of Medicine, Seoul 07061, Republic of Korea; sjhlove0812@gmail.com; 2Department of Ophthalmology, Seoul National University Hospital, Seoul 07061, Republic of Korea; 3The One Seoul Eye Clinic, Seoul 06027, Republic of Korea; xcski@hanmail.net; 4Department of Biostatistics, Seoul National University Boramae Medical Center, Seoul 07061, Republic of Korea; oh.sohee@gmail.com; 5Department of Ophthalmology, Seoul National University Boramae Medical Center, Seoul 07061, Republic of Korea

**Keywords:** phacotrabeculectomy, glaucoma, cataract surgery, prediction error, refraction

## Abstract

Purpose: To compare refractive prediction errors between phacotrabeculectomy and phacoemulsification. Methods: Refractive prediction error was defined as the difference in spherical equivalent between the predicted value using the Barrett Universal II formula and the actual value obtained at postoperative one month. Forty-eight eyes that had undergone phacotrabeculectomy (19 eyes, open-angle glaucoma; 29 eyes, angle-closure glaucoma) were matched with 48 eyes that had undergone phacoemulsification by age, average keratometry value and axial length (AL), and their prediction errors were compared. The factors associated with prediction errors were analyzed by multivariable regression analyses. Results: The phacotrabeculectomy group showed a larger absolute prediction error than the phacoemulsification group (0.51 ± 0.37 Diopters vs. 0.38 ± 0.22 Diopters, *p* = 0.033). Larger absolute prediction error was associated with longer AL (*p* = 0.010) and higher intraocular pressure (IOP) difference (*p* = 0.012). Hyperopic shift (prediction error > 0) was associated with shallower preoperative anterior chamber depth (ACD) (*p* = 0.024) and larger IOP difference (*p* = 0.031). In the phacotrabeculectomy group, the prediction error was inversely correlated with AL: long eyes showed myopic shift and short eyes hyperopic shift (*p* = 0.002). Conclusions: Surgeons should be aware of the possibility of worse refractive outcomes when planning phacotrabeculectomy, especially in eyes with high preoperative IOP, shallow ACD, and/or extreme AL.

## 1. Introduction

Glaucoma and cataract are major and often coexisting causes of visual impairment in the elderly [1]. When these diseases require surgical intervention, clinicians have to decide whether to perform glaucoma filtration surgery and phacoemulsification at the same time or sequentially. Phacotrabeculectomy has several advantages in terms of minimizing postoperative intraocular pressure (IOP) spikes and reducing surgery time and cost, with less recovery time [2,3,4]. Even in cases with angle-closure glaucoma (ACG), where IOP can be lowered by phacoemulsification alone, phacotrabeculectomy has been reported to provide additional benefits in the aspect of cost-effectiveness [5] and IOP-lowering extent [6]. Therefore, phacotrabeculectomy can be considered to be a viable option for cases involving either open-angle glaucoma (OAG) or ACG.

As a refractive surgery, however, phacotrabeculectomy has a limitation. Trabeculectomy induces changes to ocular biometry with regard to axial length (AL) and anterior chamber depth (ACD) [7,8,9,10,11]. Such changes might increase prediction error, since intraocular lens (IOL) power calculation is based on preoperative ocular biometry. Several studies have compared refractive outcomes between phacotrabeculectomy and phacoemulsification alone [11,12,13,14,15,16,17]. The results, however, are inconsistent, and moreover, the prediction error was calculated using a third-generation IOL calculation formula, which uses only AL and keratometry values to determine the effective lens position [18]. Currently, IOL power calculation is based on fourth-generation formulae that incorporate additional variables, including ACD, in order to predict effective lens position more accurately [18,19]. This enhanced accuracy might not be relevant to phacotrabeculectomy cases though, since glaucoma patients with uncontrolled intraocular pressure (IOP) would have pathologic aqueous outflow pathway and, thus, deviated preoperative ACD values. Further, this deformation would be different between ACG and OAG patients. For purposes of clarification and confirmation, therefore, the present study compared post-phacotrabeculectomy and phacoemulsification prediction errors using a fourth-generation IOL power calculation formula (i.e., the Barrett II Universal formula) and further compared prediction errors between ACG and OAG eyes to evaluate the effect of preoperative assessment of effective lens position.

## 2. Materials and Methods

This retrospective comparative study was conducted following the tenets set forth in the Declaration of Helsinki and was approved by the Institutional Review Board of Seoul National University Boramae Medical Center (no. 20-2022-110). Between January 2009 and June 2022, glaucoma patients who had undergone uncomplicated phacotrabeculectomy performed by either of two glaucoma specialists (KML or SHK) at the Seoul National University Boramae Medical Center were included. Control cases were selected from among phacoemulsification cases (without glaucoma) performed by the same surgeons in the same study period. The control cases were matched individually to the phacotrabeculectomy cases according to age (within 2 years), average keratometry value (average K, within 1 Diopter) and axial length (AL, within 1 mm). The exclusion criteria were as follows: extracapsular bag location of IOL (sulcus or scleral fixation); previous history of ocular surgery such as corneal refractive surgery; surgical complications such as posterior capsular rupture or concomitant anterior/posterior vitrectomy; and less than one-month follow-up on refractive errors.

Preoperatively, all of the subjects underwent comprehensive ophthalmologic examinations that included best-corrected visual acuity (BCVA) assessment, refraction, slit-lamp biomicroscopy, Goldmann applanation tonometry, gonioscopy, dilated funduscopic examination, keratometry (RKT-7700; Nidek, Hiroshi, Japan) and ocular biometry (IOLMaster version 5; Carl Zeiss Meditec, Dublin, CA, USA). In all phacotrabeculectomy cases, IOP and ocular biometry were measured after maximal medical treatment to reduce IOP as much as possible. Among the ocular biometry values, ACD and AL were used in the analysis.

In general, trabeculectomy was performed through the fornix-based conjunctival flap and superior rectangular 3.5 × 2.5 mm scleral flap with intraoperative mitomycin C (0.4 mg/mL for 3 min). The scleral flap and conjunctiva were sutured with interrupted 10-0 nylon sutures. Phacoemulsification was performed through a 2.75 mm clear corneal incision at a different site from the trabeculectomy, and single-piece acrylic IOLs were placed in the bag in all cases. The corneal wound was repaired with a 10-0 nylon interrupted suture in the combined phacotrabeculectomy group and with stromal hydration in the phacoemulsification group. The postoperative IOP and refractive errors were measured at one month after surgery. ∆IOP was defined as the difference between preoperative IOP and postoperative IOP. The prediction error was obtained by subtracting the actual postoperative spherical equivalent (SE) from the predicted SE using the Barrett Universal II formula.

### Data Analysis

The intergroup comparisons (phacotrabeculectomy vs. phacoemulsification) were performed by way of independent *t*-test for the continuous variables and by chi-square testing for the categorical variables. Comparisons by diagnosis (OAG, ACG and control) were performed by analysis of variance (ANOVA) test, applying the post hoc Scheffe test for the continuous variables and the chi-square test for the categorical variables. Univariable and multivariable analyses were run to determine the factors associated with prediction errors, and parameters with a *p*-value less than 0.10 in the univariable analysis were included in the subsequent multivariable analysis. A logistic regression analysis was performed to determine the factors associated with hyperopic shift (prediction error > 0). Statistical analyses were performed with commercially available software (Stata version 16.0; StataCorp, College Station, TX, USA). The data herein are presented as mean ± standard deviations except where stated otherwise, and the cutoff for statistical significance was set to *p* < 0.05.

In calculating the required sample size, we considered that our aim was to detect prediction errors larger than 0.5 Diopters. And for an assumed standard deviation of 0.75 Diopters, a sample of 37 cases was deemed to be required to detect a difference in prediction error with 80% power using a 2-sided 5%-level independent *t*-test.

## 3. Results

A total of 53 phacotrabeculectomy eyes were enrolled between January 2009 and June 2022. Of these, one eye was excluded due to combined anterior vitrectomy due to zonular laxity, one eye due to IOL sulcus insertion, one eye due to previous history of trabeculectomy, and two eyes due to follow-up loss in the aspect of refractive errors, which resulted in a final sample of 48 eyes of 48 subjects (nineteen OAG eyes and twenty-nine ACG eyes). During the same period, 48 eyes were matched by age, average keratometry value and AL from the uneventful phacoemulsification cases without glaucoma history. The two groups’ demographic and clinical characteristics are summarized in Table 1: there were no differences in age, sex distribution, AL or average K, whereas the phacotrabeculectomy group had higher preoperative IOP (*p* < 0.001), larger ∆IOP (*p* < 0.001) and shallower preoperative ACD (*p* = 0.047). During the study period, we used two kinds of four-haptic IOLs and three kinds of two-haptic IOLs. For the four-haptic IOLs, 48 cases (50%) used CLARE (Cristalens, Paris, France), and 26 cases (27%) used Akreos-AO MI60 (Bauch & Lomb, Rochester, NY, USA). For the two-haptic IOLs, 17 cases (18%) used Tecnis ZBC100 (Johnson & Johnson Vision, Santa Ana, CA, USA), 3 cases (3%) used AcrySof SN60AF (Alcon Inc., Fort Worth, TX, USA) and 2 cases (2%) used Sensar AR40e (Johnson & Johnson Vision). The four-haptic IOLs were more frequently used in the phacotrabeculectomy group than in the phacoemulsification group (*p* = 0.004), but the prediction error did not differ, regardless of four-haptic IOL (−0.29 ± 0.45 Diopters) or two-haptic IOL (−0.20 ± 0.54 Diopters, *p* = 0.440).

The prediction error was −0.23 ± 0.59 Diopters in the phacotrabeculectomy group and −0.31 ± 0.31 in the phacoemulsification group (*p* = 0.436). The absolute prediction error (absolute value of prediction error) was larger in the phacotrabeculectomy group than in the phacoemulsification group (0.51 ± 0.37 Diopters vs. 0.38 ± 0.22 Diopters, *p* = 0.033, Table 1). The factors associated with the absolute prediction errors were analyzed by multiple regression analyses using two models, which showed that longer AL (*p* = 0.010 in model 1, *p* = 0.020 in model 2), larger ∆IOP (*p* = 0.012) and phacotrabeculectomy (*p* = 0.025) were associated with larger absolute prediction error (Table 2). This result was not changed when preoperative IOP was used instead of ∆IOP (Appendix A).

Hyperopic shift of prediction error was observed in 23 (24%) cases: 15 phacotrabeculectomy cases and 8 phacoemulsification cases (*p* = 0.094). The logistic regression analyses revealed that shallower preoperative ACD (*p* = 0.024) and larger ∆IOP (*p* = 0.004 in model 1, *p* = 0.031 in model 2) were associated with higher odds of hyperopic shift (Table 3). This result was not changed when preoperative IOP was used instead of ∆IOP (Appendix A).

The phacotrabeculectomy group consisted of OAG and ACG subgroups. The AL was longer in the OAG subgroup than in the control or ACG subgroup (24.6 ± 1.7 mm vs. 23.6 ± 1.6 mm vs. 22.8 ± 0.7 mm, *p* < 0.001), whereas the preoperative ACD was shallower in ACG than in the control or OAG (2.43 ± 0.31 mm vs. 2.99 ± 0.54 mm vs. 3.28 ± 0.38 mm, *p* < 0.001). ∆IOP was on the (highest to lowest) order of ACG, OAG and control group (21.1 ± 12.7 mmHg vs. 9.5 ± 6.7 mmHg vs. 2.1 ± 2.6 mmHg, *p* < 0.001). The prediction errors were −0.41 ± 0.51 Diopters in OAG and −0.12 ± 0.62 in ACG, neither of which statistically differed from the control group (*p* = 0.077). In the multiple regression analysis, the prediction error showed a statistically significant correlation with AL (*p* = 0.002) and a marginally significant correlation with preoperative ACD (*p* = 0.056, Table 4, Figure 1, Appendix A). This result was not changed when preoperative IOP was used instead of ∆IOP (Appendix A).

## 4. Discussion

In this study, we discovered that absolute prediction error was larger in the phacotrabeculectomy group than in the phacoemulsification group. Longer AL and larger IOP change were the risk factors for higher absolute prediction error. Hyperopic shift (=positive value of prediction error) was associated with shallower preoperative ACD and larger IOP change, both of which were more common in ACG eyes. In the phacotrabeculectomy group, the prediction error showed an inverse correlation with AL, which means that eyes with long AL showed myopic shift, while eyes with short AL showed hyperopic shift. To the best of our knowledge, this study is the first to compare prediction errors after phacotrabeculectomy and phacoemulsification using the Barrett Universal II formula, which incorporates both ACD and AL measured preoperatively.

Final refractive errors can be either myopic (=negative value of prediction error) or hyperopic (=positive value of prediction error) by prediction. The phacotrabeculectomy group had larger deviations from the prediction in both myopic and hyperopic shift cases, and thus, averaging them would have concealed the actual refractive errors; this is why we used the absolute value of prediction error. Several studies also have reported worse refractive outcomes when phacoemulsification and trabeculectomy were performed simultaneously [12,13,15,17]. In the current study, AL and larger IOP change were both associated with larger prediction error. We speculated that the final refractive error would be affected more in the phacotrabeculectomy group, since AL and IOP presumably would have changed more after phacotrabeculectomy.

AL is known to be shortened after either trabeculectomy [7,9,10,20] or phacotrabeculectomy [11], and the extent of AL decrease is correlated with the amount of IOP reduction [7,10,11,20]. Although AL shortening was also reported after phacoemulsification alone, AL shortening was more prominent after phacotrabeculectomy in the study in question [11]. Accordingly, postoperative AL would deviate more from preoperative AL in phacotrabeculectomy than in phacoemulsification. Since the IOL power calculation formulae were developed based on phacoemulsification data, their predictions would be less accurate in phacotrabeculectomy cases. Notably, Lee et al. reported a post-trabeculectomy association between prediction error and IOP change [13]. We speculated that large IOP change would be associated with greater AL shortening, which would result in more prediction error in phacotrabeculectomy.

Several studies, however, have reported that prediction error did not differ between phacotrabeculectomy and phacoemulsification [11,14,16]. These studies calculated prediction error using the SRK II formula (if AL ≤ 26 mm) and SRK/T formula (if AL > 26 mm) [14] or the average of the SRK/T, Holladay 1 and Hoffer Q formulae [11]. As third-generation formulae, all of them (SRK/T, SRK II, Holladay 1 and Hoffer Q) use only AL and keratometry values in their IOL power calculation, under the assumption that the effective lens position is directly related to AL [18,21]. We speculated that the nonapplication of ACD might have been the reason for the lack of intergroup difference in these studies. In the OAG eyes with high preoperative IOP, the preoperative ACD might have been overmeasured due to stasis of aqueous humor in the anterior chamber. If so, the actual effective lens position after phacotrabeculectomy would have been more anteriorly located than the effective lens position calculated from the preoperative ACD, thereby resulting in myopic shift. Contrastingly, in ACG eyes with shallow ACD, the actual effective lens position after phacotrabeculectomy would be more posteriorly located due to anterior chamber deepening, thus resulting in hyperopic shift. To summarize, prediction errors might occur even with the use of fourth-generation formulae, since these measure not only the AL but also many other parameters, including ACD, for more accurate determination of effective lens position.

Prediction error also is dependent on the applied formula, since it is well known that the accuracy of each formula is dependent on the AL range [21,22]. For consistency of comparison, however, it would be better to use a singular IOL calculation formula. The superiority of the Barrett Universal II formula over SRK/T formula was proven in phacotrabeculectomy cases [23]. Thus, we used the Barrett Universal II formula, which is considered to be one of the best options for covering the entire AL range [19,24]. By this means, we demonstrated worse refractive outcomes in the phacotrabeculectomy group over the entire range of AL.

Interestingly, our phacoemulsification group also showed a myopic shift of −0.3 Diopters. One possible explanation is twofold. First, AL shortening also was noted after phacoemulsification only [11]. This change may vary among individuals, since it might be associated with tissue properties, as supported by a previous study showing that eyes with differing corneal hysteresis showed differing AL shortening after trabeculectomy [25]. Thus, comparison of populations with different tissue properties would lead to slight myopic shifts from predictions. The second part of our tentative explanation for the myopic shift in the phacoemulsification group is the difference in crystalline lens anatomy among individuals. The geometric center of the crystalline lens is located anteriorly to the exact half point along the lens thickness; that is, there is more convexity to the posterior side than to the anterior side [26]. Thus, the effective lens position might be also located more anteriorly than the calculated position after phacoemulsification. Moreover, this anterior/posterior difference in lens shape is affected by both thickness and age [26]. Further study accounting for these factors would be helpful in order to reduce refractive errors after either phacotrabeculectomy or phacoemulsification.

This study has several limitations. First, the sample size was relatively small and not evenly distributed along the whole range of AL. Second, due to the retrospective nature of the study, the applied treatment protocol in the phacotrabeculectomy group (e.g., preoperative antiglaucoma medications) might have differed among patients. Third, we were unable to obtain postoperative AL and ACD for comparison. Further prospective longitudinal study would be helpful in order to evaluate the effect of ocular biometry changes on final refractive outcomes after phacotrabeculectomy. Fourth, the follow-up period was short: the final refractive errors and postoperative IOP values were determined at postoperative one month. Although several studies have reported that refractive error was stabilized one week after phacoemulsification [27,28], it is unknown whether the same would be true after phacotrabeculectomy [29,30]. Some post-phacotrabeculectomy patients have shown unstable refractive errors when followed-up for longer periods [29]. Chung et al., however, reported nondifference in long-term refractive outcomes between phacotrabeculectomy and phacoemulsification groups [30]. Nevertheless, it should be noted that refractive error could change with IOP change, especially in glaucoma patients who have undergone phacotrabeculectomy and been followed-up for longer, particularly those with IOP fluctuation. Fifth, and finally, we were unable to suggest a better way to minimize prediction error. Simply, we identified patients at high risk of worse refractive outcome. Moreover, a portion of the prediction error might originate from diseases themselves (such as ACG) rather than from phacotrabeculectomy. Our results, however, may nonetheless be helpful to clinicians’ surgery planning, specifically by alerting them to patients at high risk of worse refractive outcomes, regardless of the particular reasons. Future study should focus on such patients in order to find a means of achieving better refractive outcomes.

In conclusion, refractive prediction error was larger in phacotrabeculectomy than in phacoemulsification cases. Such inaccuracies may accrue from AL and IOP changes, both of which resulted in changes in effective lens position. Surgeons, therefore, should be aware of the possibility of worse refractive outcomes when planning phacotrabeculectomy in eyes with high preoperative IOP, shallow ACD and/or extreme AL.

## Figures and Tables

**Figure 1 jcm-12-05706-f001:**
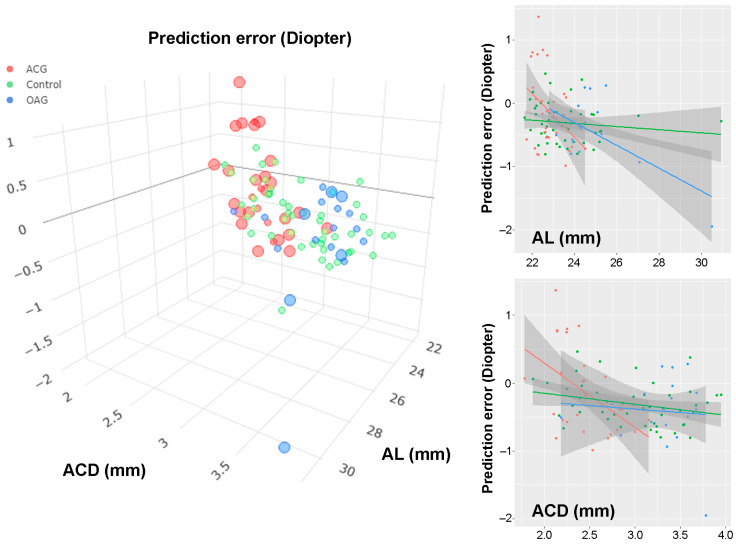
Scatter plot of prediction errors (based on the Barrett Universal ll formula) according to axial length (AL) and anterior chamber depth (ACD): The red dots indicate angle-closure glaucoma (ACG) patients, the blue dots open-angle glaucoma (OAG) patients and the green dots control patients who underwent phacoemulsification only. In the three-dimensional plot (left), intraocular pressure (IOP) change is marked by sphere size: large sphere, ∆IOP ≥ 10 mmHg; small sphere, ∆IOP ≤ 10 mmHg. The regression lines are drawn for AL (right top) and ACD (right bottom). Please note the larger prediction error in the OAG and ACG patients who underwent phacotrabeculectomy: OAG patients with long AL showed myopic shift, while ACG patients with short ACD showed hyperopic shift. Prediction error was larger in cases with remarkable IOP change.

**Table 1 jcm-12-05706-t001:** Demographic comparison.

	Phacotrabeculectomy (*n* = 48)	Phacoemulsification (*n* = 48)	*p* Value
Age, years	66.1 ± 11.1	68.9 ± 10.0	0.198 ^a^
Sex (Male/Female)	19/29	20/28	0.835 ^b^
AL, mm (range)	23.5 ± 1.5 (21.64–30.92)	23.6 ± 1.6 (21.73–30.47)	0.814 ^a^
ACD, mm	2.77 ± 0.54	2.99 ± 0.54	0.047 ^a^
Average K, Diopter	44.3 ± 1.6	44.4 ± 1.4	0.722 ^a^
IOL type (four-haptic/two-haptic)	17/31	5/43	0.004
Preoperative IOP, mmHg	27.6 ± 11.5	13.0 ± 3.9	<0.001 ^a^
∆IOP, mmHg	16.5 ± 12.1	2.1 ± 2.6	<0.001 ^a^
Prediction error, Diopter	−0.23 ± 0.59	−0.31 ± 0.31	0.436
Absolute prediction error, Diopter	0.51 ± 0.37	0.38 ± 0.22	0.033
Diagnosis		OAG (39.6%) ACG (60.4%)	

AL = axial length; ACD = anterior chamber depth; IOL = intraocular lens; IOP = intraocular pressure; OAG = open-angle glaucoma; ACG = angle-closure glaucoma. ^a^ Comparison performed using independent *t*-test; ^b^ Comparison performed using chi-square test.

**Table 2 jcm-12-05706-t002:** Factors associated with absolute prediction error.

	Univariable Analysis			Multivariable Analysis (Model 1) ^a^			Multivariable Analysis (Model 2) ^a^		
	Coefficient	95% CI	*p*	Coefficient	95% CI	*p*	Coefficient	95% CI	*p*
Age, years	−0.001	−0.007, 0.005	0.765						
Female (vs. male sex)	−0.062	−0.190, 0.066	0.341						
**AL, mm**	**0.046**	**0.006, 0.087**	**0.026**	**0.053**	**0.013, 0.092**	**0.010**	**0.047**	**0.008, 0.087**	**0.020**
ACD, mm	0.031	−0.086, 0.148	0.598						
Average K, Diopter	−0.009	−0.051, 0.034	0.694						
Four-haptic IOL (vs. two-haptic IOL)	0.056	−0.094, 0.206	0.460						
∆ **IOP, mmHg**	**0.006**	**0.001, 0.012**	**0.030**	**0.007**	**0.002, 0.012**	**0.012**			
**Phacotrabeculectomy (vs. control group)**	**0.135**	**0.011, 0.258**	**0.033**				**0.138**	**0.018, 0.259**	**0.025**

CI = confidence interval; AL = axial length; ACD = anterior chamber depth; IOL = intraocular lens; IOP = intraocular pressure; ^a^ Variables with *p* < 0.10 in the univariable analysis were included in the subsequent multivariable analysis. Owing to multicollinearity between the groups and ∆IOP, two multivariable analysis models were constructed. The result of multivariable analysis with backward elimination is equivalent to that of Model 1. Statistically significant values (*p* < 0.05) are shown in bold.

**Table 3 jcm-12-05706-t003:** Risk factors for hyperopic prediction error.

	Univariable Analysis			Multivariable Analysis (Model 1) ^a^			Multivariable Analysis (Model 2) ^a^		
	OR	95% CI	*p*	OR	95% CI	*p*	OR	95% CI	*p*
Age, years	1.030	0.982, 1.081	0.220						
Female (vs. male sex)	1.784	0.655, 4.856	0.257						
AL, mm	0.570	0.341, 0.953	0.032	0.647	0.385, 1.086	0.100			
**ACD,** **mm**	**0.180**	**0.062, 0.517**	**0.001**				**0.284**	**0.093, 0.868**	**0.027**
Average K, Diopter	1.186	0.862, 1.631	0.296						
Four-haptic IOL (vs. two-haptic IOL)	0.763	0.540, 1.079	0.126						
∆ **IOP, mmHg**	**1.072**	**1.028, 1.118**	**0.001**	**1.077**	**1.013, 1.144**	**0.017**	**1.062**	**0.998, 1.130**	**0.057**
Phacotrabeculectomy (vs. control group)	0.821	−0.153, 1.795	0.099	0.709	0.167, 3.001	0.641	0.690	0.162, 2.941	0.616

CI = confidence interval; AL = axial length; ACD = anterior chamber depth; IOL = intraocular lens; IOP = intraocular pressure; ^a^ Variables with *p* < 0.10 in the univariable analysis were included in the subsequent multivariable analysis. Because of multicollinearity between AL and ACD, two models were constructed. The result of multivariable analysis with backward elimination is equivalent to that of Model 2. Statistically significant values (*p* < 0.05) are shown in bold.

**Table 4 jcm-12-05706-t004:** Factors associated with prediction error after phacotrabeculectomy.

	Univariable Analysis			Multivariable Analysis (Model 1) ^a^			Multivariable Analysis (Model 2) ^a^		
	Coefficient	95% CI	*p*	Coefficient	95% CI	*p*	Coefficient	95% CI	*p*
Age, years	0.007	−0.008, 0.023	0.359						
Female (vs. male sex)	0.402	0.068, 0.736	0.019	0.146	−0.206, 0.498	0.407	0.296	−0.063, 0.654	0.103
**AL, mm**	**−0.195**	**−0.297, −0.094**	**<0.001**	**−0.213**	**−0.346, −0.080**	**0.002**			
ACD, mm	−0.418	−0.720, −0.115	0.008				−0.472	−0.957, 0.013	0.056
Average K, Diopter	0.064	−0.043, 0.171	0.236						
Four-haptic IOL (vs. two-haptic IOL)	0.054	−0.513, 0.622	0.848						
∆IOP, mmHg	0.012	−0.002, 0.026	0.084	0.011	−0.004, 0026	0.133	0.005	−0.011, 0.020	0.556
ACG (vs. OAG)	0.294	−0.049, 0.638	0.091	−0.277	−0.709, 0.155	0.203	−0.273	−0.814, 0.268	0.315

CI = confidence interval; AL = axial length; ACD = anterior chamber depth; IOL = intraocular lens; IOP = intraocular pressure; ACG = angle-closure glaucoma; OAG = open-angle glaucoma; ^a^ Variables with *p* < 0.10 in the univariable analysis were included in the subsequent multivariable analysis. Because of multicollinearity between AL and ACD, two models were constructed. Multivariable analysis with backward elimination left only AL as the associative factor. Statistically significant values (*p* < 0.05) are shown in bold.

## Data Availability

The data generated during this study are available from the corresponding author on reasonable request.

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
