# Peer review of "Factors Associated with Refractive Prediction Error after Phacotrabeculectomy"

_jcm, 2023, doi:10.3390/jcm12175706_

Round 1
Reviewer 1 Report
The manuscript submitted by the authors presents an original study concerning refractive error following phacotrabeculectomy and phacoemulsification. The authors chose to work with the Barrett Universal II formula. A higher refractive prediction error was found after phacotrabeculectomy than after phacoemulsification. This error was correlated with axial length (long AL), anterior chamber depth (shallow ACD), and IOP change (larger IOP delta).
The article is well-written and engaging for the reader, with a proper structure. The methods and results are described clearly. A control group was chosen and described correctly. The discussion is thorough. However, the article lacks a "Conclusions" section, which I suggest adding.
Major concerns: None
Minor concerns:
Line 19: The PE in the phacoemulsification group (0.38 ± 0.22) lacks the unit 'Diopters.'
The introduction is rather concise. I encourage the authors to expand this section, to provide broader insight into the topic. E.g., Line 35: In what terms was phacotrabeculectomy reported as 'better' than phacoemulsification? Etc.
Line 79: As the time span for incorporation into the study is rather long (2009-2022), it is possible that various types of IOLs may have been implanted. Please provide information on the types of IOL/IOLs. Is there a risk that the type of IOL was a factor predisposing the refractive prediction error?
Line 84: When was the postoperative IOP measured in the study and control groups? It tends to fluctuate even within a few months after trabeculectomy.
Table 1: The standard deviation for AL is rather small in both groups; can you provide the range?
Tables 2 and 3: Do these data include both the study and control groups or only the study group? Why is the type of intervention (phacotrabeculectomy vs. phacoemulsification) considered only in Table 2, and not in Table 3? Can you include the initial IOP in the analysis, not only the IOP change?
Line 147: Showed statistically significant and marginally significant CORRELATION.
Table 4: Shouldn't the p-values for sex (0.019), ACD (0.008 and 0.056) also be in bold?
Line 164: There is a double dot present; please correct.
Line 203: Correct the typo: 'non-applicatoin'
Author Response
Thank you very much for your comments. We are really appreciative.
1. Line 19: The PE in the phacoemulsification group (0.38 ± 0.22) lacks the unit 'Diopters.'
Thank you very much. We have corrected it (page 1 line 19 and page 4 line 138).
2. The introduction is rather concise. I encourage the authors to expand this section, to provide broader insight into the topic. E.g., Line 35: In what terms was phacotrabeculectomy reported as 'better' than phacoemulsification? Etc.
We totally agree with you. Inspired by this great suggestion, we have revised the Introduction as follows (page 1 lines 32–38 and page 2 lines 52–57):
“Phacotrabeculectomy has several advantages in terms of minimizing postoperative intraocular pressure (IOP) spikes and reducing surgery time and cost, with less recovery time.2-4 Even in cases with angle-closure glaucoma (ACG), where IOP can be lowered by phacoemulsification alone, phacotrabeculectomy has been reported to provide additional benefits in the aspect of cost-effectiveness5 and IOP-lowering extent.6 Therefore, phacotrabeculectomy can be considered to be a viable option for cases involving either open-angle glaucoma (OAG) or ACG.”
“For purposes of clarification and confirmation, therefore, the present study compared post-phacotrabeculectomy and phacoemulsification prediction errors using a fourth-generation IOL power calculation formula (i.e., the Barrett II Universal formula), and further compared prediction errors between ACG and OAG eyes to evaluate the effect of preoperative assessment of the effective-lens position.”
3. Line 79: As the time span for incorporation into the study is rather long (2009-2022), it is possible that various types of IOLs may have been implanted. Please provide information on the types of IOL/IOLs. Is there a risk that the type of IOL was a factor predisposing the refractive prediction error?
This is a very good point! During the study period, we used one-piece hydrophilic acrylic square-edged IOL (CLARE; Cristalens, Paris, France) in 48 cases (50%), one-piece hydrophobic acrylic square-edged IOL (Akreos-AO MI60; Bauch & Lomb, Rochester, NY, USA) in 26 cases (27%), one-piece hydrophobic IOL (Tecnis ZCB100; Johnson & Johnson Vision, Santa Ana, CA, USA) in 17 cases (18%), another one-piece hydrophobic IOL (AcrySof SN60AF; Alcon Inc, Fort Worth, TX, USA) in 3 cases (3%), and three-piece IOL (Sensar AR40e; Johnson & Johnson Vision) in 2 cases (2%). One-piece acrylic square-edged IOL with four haptics (CLARE and Akreos-AO) were more frequently used in the phacotrabeculectomy group (43 cases, 90%) than in the phacoemulsification group (31 cases, 65%). Refractive error, however, did not differ by IOL type (four-haptic IOL vs. two-haptic IOL): -0.29 ± 0.45 vs. -0.20 ± 0.54 (P=0.440, independent-test). Inspired by your brilliant suggestion, the information on the IOL type is now clarified in the Results (page 3 lines 122–130) as follows:
“During the study period, we used two kinds of four-haptic IOLs and three kinds of two-haptic IOLs. For the four-haptic IOLs, 48 cases (50%) used CLARE (Cristalens, Paris, France), and 26 cases (27%) used Akreos-AO MI60 (Bauch & Lomb, Rochester, NY, USA). For the two-haptic IOLs, 17 cases (18%) used Tecnis ZBC100 (Johnson & Johnson Vision, Santa Ana, CA, USA), 3 cases (3%) used AcrySof SN60AF (Alcon Inc., Fort Worth, TX, USA), and 2 cases (2%) used Sensar AR40e (Johnson & Johnson Vision). The four-haptic IOLs were more frequently used in the phacotrabeculectomy group than in the phacoemulsification group (P=0.004), but the prediction error did not differ regardless of four-haptic IOL (-0.29 ± 0.45 Diopters) or two-haptic IOL (-0.20 ± 0.54 Diopters, P=0.440).”
Further, we clarified the types of IOL in Table 1, and the IOL type was included in all of the following analyses (Tables 2–4). Your brilliant idea has improved our manuscript. Thank you very much.
4. Line 84: When was the postoperative IOP measured in the study and control groups? It tends to fluctuate even within a few months after trabeculectomy.
As we had stated in the Methods, we measured postoperative IOP at one month after surgery (page 2 lines 88–89). As the reviewer pointed out, IOP might fluctuate after glaucoma filtration surgery, but its fluctuation is within a much smaller range compared with perioperative IOP change. The non-difference of long-term refractive outcomes between the phacotrabeculectomy and phacoemulsification groups supports this speculation (Ref #30). The reviewer’s point, however, is a very important one, and we have corrected the limitations section as follows (page 9 lines 267–277):
“Fourth, the follow-up period was short: the final refractive errors and postoperative IOP values were determined at postoperative one month. Although several studies have reported that refractive error was stabilized one week after phacoemulsification,27, 28 it is unknown whether the same would be true after phacotrabeculectomy.29, 30 Some post-phacotrabeculectomy patients have shown unstable refractive errors when followed up for longer periods.29 Chung et al., however, reported non-difference of long-term refractive outcomes between phacotrabeculectomy and phacoemulsification groups.30 Nevertheless, it should be noted that refractive error could change with IOP change, especially in glaucoma patients who had undergone phacotrabeculectomy and been followed up longer, particularly those with IOP fluctuation.”
5. Table 1: The standard deviation for AL is rather small in both groups; can you provide the range?
Yes, the ranges have been updated in Table 1.
6. Tables 2 and 3: Do these data include both the study and control groups or only the study group? Why is the type of intervention (phacotrabeculectomy vs. phacoemulsification) considered only in Table 2, and not in Table 3? Can you include the initial IOP in the analysis, not only the IOP change?
This is a very important point! We have corrected Table 3 to include the type of intervention. IOP change is largely dependent on the initial IOP; thus, for the analyses, we selected IOP change instead of initial IOP. According to the reviewer’s suggestion, we have added Supplemental Tables 1–3, which include preoperative IOP instead of IOP change, and revised the Results (page 4 lines 142–143, page 5 line 155, and page 6 lines 170–171). Thank you very much.
7. Line 147: Showed statistically significant and marginally significant CORRELATION.
We have corrected it according to your suggestion (page 6 lines 169–170). Thank you.
8. Table 4: Shouldn't the p-values for sex (0.019), ACD (0.008 and 0.056) also be in bold?
No, because the P-value for sex was not less than 0.05 in the multivariable analysis, and that for ACD was larger than 0.05.
9. Line 164: There is a double dot present; please correct.
Yes, a double dot is visible at the end of the Figure 1 legend in the pdf version, but not in our original paper. We will ask the editors to correct this typo. Thank you very much.
10. Line 203: Correct the typo: 'non-applicatoin'
The typo was corrected (page 8 line 226). Thank you very much for your wonderful comments, which have greatly improved our study.
Reviewer 2 Report
Thank you for asking me to review this manuscript. The authors conducted a retrospective study to assess the factors associated with refractive prediction error after phacotrabeculectomy. Overall, the manuscript is interesting, however, I have few questions about some details related to the interpretation of the results.
First, power and sample size calculation should be added. The authors mentioned in the limitation that the sample size was relatively small. However, sample size should be calculated during study design to ensure sufficient power.
The second one is about the selection of control cases. It mentioned in the manuscript that control cases were matched by age, average K and AL. Although table 1 showed that the factors were similar in the two groups, it will be much clearer if the authors can provide more details of matching procedure (e.g., the matching method).
Third, 48 eyes of 48 subjects (19 OAG eyes and 29 ACG eyes) were included in the analysis. Whether different types of glaucoma affect treatment outcomes and whether eyes from right/left also need to be considered. According to the scatter plots in Figure 1, the slope trends of OAG and ACG seem to be different.
May need minor editing of English language.
Author Response
Thank you very much for your comments. We are really appreciative.
1. First, power and sample size calculation should be added. The authors mentioned in the limitation that the sample size was relatively small. However, sample size should be calculated during study design to ensure sufficient power.
This is a very important point! We have added the sample size calculation to the Methods as follows (page 3 lines 107–110):
“In calculating the required sample size, we considered that our aim was to detect prediction error larger than 0.5 Diopters. And for an assumed standard deviation of 0.75 Diopters, a sample of 37 cases was deemed to be required to detect a difference in prediction error with 80% power using a 2-sided 5%-level independent t-test.”
2. The second one is about the selection of control cases. It mentioned in the manuscript that control cases were matched by age, average K and AL. Although table 1 showed that the factors were similar in the two groups, it will be much clearer if the authors can provide more details of matching procedure (e.g., the matching method).
This is another very important point! We have clarified the matching procedure as follows (page 2 lines 64–68):
“Control cases were selected from among phacoemulsification cases (without glaucoma) performed by the same surgeons in the same study period. The control cases were matched individually to the phacotrabeculectomy cases according to age (within 2 years), average keratometry value (average K, within 1 Diopter), and axial length (AL, within 1 mm).”
3. Third, 48 eyes of 48 subjects (19 OAG eyes and 29 ACG eyes) were included in the analysis. Whether different types of glaucoma affect treatment outcomes and whether eyes from right/left also need to be considered. According to the scatter plots in Figure 1, the slope trends of OAG and ACG seem to be different.
As we have already stated in the limitations section, we also think that the difference in the applied treatment protocol is a major limitation of this study (page 9 lines 264–266). One of the major findings of this study, however, is the difference of predictive error type after phacotrabeculectomy between OAG and ACG eyes: myopic shift in OAG, and hyperopic shift in ACG. As we stated in the Discussion (page 8 lines 222–239), the estimation of effective lens position would be incorrect in the case of uncontrolled glaucoma. To demonstrate this more clearly, we have revised the Introduction as follows (page 2 lines 52–57):
“Further, this deformation would be different between ACG and OAG patients. For purposes of clarification and confirmation, therefore, the present study compared post-phacotrabeculectomy and phacoemulsification prediction errors using a fourth-generation IOL power calculation formula (i.e., the Barrett II Universal formula), and further compared prediction errors between ACG and OAG eyes to evaluate the effect of preoperative assessment of the effective-lens position.”
We do not think that laterality of eyes would matter. The prediction error did not differ between the right and left eyes (-0.23 ± 0.47 vs. -0.32 ± 0.48, P=0.345, independent t-test). Also, the slopes did not differ according to diagnosis when evaluated by Fisher’s Z-test with Bonferroni correction.
|
Prediction error & ACD |
Pearson correlation |
Spearman correlation |
Fisher’s z-test (Pearson correlation) |
Fisher’s z-test (Spearman correlation) |
|
Control |
-0.279 |
-0.264 |
Vs. OAG P=0.471 |
Vs. OAG P=0.288 |
|
OAG |
-0.077 |
0.039 |
Vs. ACG P=0.172 |
Vs. ACG P=0.100 |
|
ACG |
-0.471 |
-0.450 |
Vs. Control P=0.364 |
Vs. Control P=0.385 |
|
Prediction error & AL |
Pearson correlation |
Spearman correlation |
Fisher’s z-test (Pearson correlation) |
Fisher’s z-test (Spearman correlation) |
|
Control |
-0.126 |
-0.209 |
Vs. OAG P=0.040 |
Vs. OAG P=0.548 |
|
OAG |
-0.620 |
-0.037 |
Vs. ACG P=0.318 |
Vs. ACG P=0.364 |
|
ACG |
-0.387 |
-0.314 |
Vs. Control P=0.253 |
Vs. Control P=0.646 |
Thank you for your wonderful comments, which have improved our manuscript a lot.
Round 2
Reviewer 2 Report
Congratulations to the authors. I have no further comments, thank you.